

SciPost Phys. Comm. Rep. 6 (2025)

# LHC EFT WG note: Basis for anomalous quartic gauge couplings

**Gauthier Durieux[1], Grant N. Remmen[2], Nicholas L. Rodd[3,4] (editors), Oscar J. P. Éboli[5], Maria C. Gonzalez-Garcia[6,7], Dan Kondo[8], Hitoshi Murayama[3,4,8] and Risshin Okabe[8]**

**1** Centre for Cosmology, Particle Physics and Phenomenology,
Université catholique de Louvain, 1348 Louvain-la-Neuve, Belgium
**2** Center for Cosmology and Particle Physics, Department of Physics,
New York University, New York, NY 10003, USA
**3** Theory Group, Lawrence Berkeley National Laboratory, Berkeley, CA 94720, USA
**4** Berkeley Center for Theoretical Physics, University of California, Berkeley, CA 94720, USA
**5** Instituto de Física, Universidade de São Paulo, São Paulo – São Paulo 05580-090, Brazil
**6** C.N. Yang Institute for Theoretical Physics, Stony Brook University,
Stony Brook, NY 11794-3849, USA
**7** Institució Catalana de Recerca i Estudis Avançats (ICREA),
Passeig Lluis Companys 23, 08010 Barcelona, Spain
**8** Kavli Institute for the Physics and Mathematics of the Universe (WPI),
The University of Tokyo, Kashiwa, Chiba 277-8583, Japan

## Abstract

**In this note, we give a definitive basis for the dimension-eight operators leading to quartic—but no cubic—interactions among electroweak gauge bosons. These are often called anomalous quartic gauge couplings, or aQGCs. We distinguish in particular the CP-even ones from their CP-odd counterparts.**

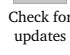

## 1 Basis

We consider the effective field theory of the standard model (SMEFT) at mass dimension eight, in particular, operators built out of at least four Higgs and electroweak gauge bosons, generating anomalous quartic gauge couplings (aQGCs). We denote operators that are CP-even as $\mathcal{O}_i$ and CP-odd ones as $\mathcal{O}\!\!\!/_i$. The basis contains operators quartic in the Higgs (S-type), bi-quadratic in the Higgs and gauge field strengths (M-type), and quartic in the gauge field strengths (T-type). The full CP-even aQGC basis is given by the $\mathcal{O}_i^S$, $\mathcal{O}_i^M$, and $\mathcal{O}_i^T$ operators in Table 1. For completeness, the full basis of CP-odd terms is given by the $\mathcal{O}\!\!\!/_i^M$ and $\mathcal{O}\!\!\!/_i^T$ operators in Table 2. There are no S-type CP-odd terms.

Regarding conventions, we use square brackets to indicate contraction of fundamental SU(2) indices and use capitalized Latin script for adjoint indices. The SU(2) generators are related to the Pauli matrices via $\tau^I = \sigma^I/2$, dual field strengths are defined as $\widetilde{B}^{\mu\nu} = \epsilon^{\mu\nu\rho\sigma}B_{\rho\sigma}/2$ and $\widetilde{W}^{I\mu\nu} = \epsilon^{\mu\nu\rho\sigma}W_{\rho\sigma}^I/2$, and we take $D_\mu H = (\partial_\mu - ig W_\mu^I \tau^I - \frac{1}{2}ig' B_\mu)H$.

## 2   Literature comparison

A brief comparison with the literature is useful. To our knowledge, no complete basis of aQGC operators with CP properties correctly identified has previously appeared. In this section, we focus on the CP-even sector of the aQGC basis. The basis presented by Almeida, Éboli, Gonzalez-Garcia, and Mizukoshi in Refs. [1–3] lists operators that are both C-even and P-even and does not include two operators that are C-odd and P-odd, but CP-even. One is of $(DH)^2BW$ form and was identified in in Ref. [4]. Similarly, Ref. [3] contains only three $(DH)^2W^2$ operators, though in fact there are four independent such terms in the CP-even basis. (Ref. [1] contained a different fourth operator, but it was found to be redundant with the three others in Ref. [2].) The original basis of T-type CP-even operators presented in Refs. [1, 2], all of which are both C-even and P-even, was incomplete. This was corrected first in Ref. [4], and subsequently in the updated basis of Ref. [3], which agrees.

The counting of operators, of both CP-even and CP-odd type, agrees with the Hilbert series analysis of Ref. [5] by Kondo, Murayama, and Okabe. However, the identification of which specific operators are CP-even and -odd in that paper contains an error. While Ref. [5] identifies $\mathcal{O}_9^M$ as CP-odd and $\mathcal{\emptyset}_6^M$ as CP-even, Ref. [4] counts them both as CP-odd. Both being P-odd, in fact $\mathcal{\emptyset}_6^M$ is C-even, and $\mathcal{O}_9^M$ is C-odd. The result is that $\mathcal{O}_9^M$ is CP-even and $\mathcal{\emptyset}_6^M$ is CP-odd. Similarly, Ref. [4] misidentified $\mathcal{\emptyset}_3^M$ as CP-even and $\mathcal{O}_8^M$ as CP-odd. Thus, while Ref. [4] contained all the operators, it had two errors associated with the CP transformation properties. The presence of a single error was noted in Ref. [5], although the wrong operator was identified in that paper.

In Table 1, we further outline the correspondence of our basis with the lists of CP-even operators in Refs. [3], [4], and [6]; see also Ref. [7]. (Note that Ref. [6] denotes the Pauli matrices as $\tau^I$.) The basis choices and notation of Refs. [4] and [6] explicitly identify the field content of each operator and, via the use of dual field strengths, cleanly categorize operators by their polarization structure. However, where possible, in this note we have chosen to follow the widely used conventions of Ref. [3], for the sake of compatibility with earlier simulations and experimental bounds. That is, we choose to introduce a minimal completion of Ref. [3] to a full basis of CP-even operators. In any case, for maximum utility and convenience, we provide a complete dictionary among all of these different conventions. The correspondence in the CP-odd case (not considered in Ref. [3]) is provided in Table 2.

In this note, we do not discuss operators at mass dimension six, for which the full basis has been identified elsewhere [8]. A useful endeavor, beyond the scope of this note, would be to identify the larger basis of operators in the Higgs effective field theory (HEFT). We leave such considerations to future work.

## 3   C and P

Given the subtleties associated with CP properties, we briefly review these details here.

Consider parity first. This is very straightforward. The Higgs is a (parity even) scalar, and derivatives transform as $f(\mu)$, where $f(\mu) = -1 + 2\delta_{\mu 0}$ equals 1 for $\mu = 0$ and $-1$ otherwise, so that the SU(2) gauge bosons transform as

$$
\begin{aligned}
\text{P}: \quad & W_{\mu\nu}^I \to f(\mu)f(\nu)\, W_{\mu\nu}^I, \\
\text{P}: \quad & \widetilde{W}_{\mu\nu}^I \to -f(\mu)f(\nu)\, \widetilde{W}_{\mu\nu}^I,
\end{aligned}
\tag{1}
$$

where the sign in the second case arises from P: $\epsilon^{\mu\nu\rho\sigma} \to -\epsilon^{\mu\nu\rho\sigma}$ (although note that $\epsilon^{IJK}$ will not flip sign, as parity is a spacetime transformation). An identical result holds for the

Table 1: The basis of CP-even aQGC operators and their C and P transformation properties. We further outline the map between our basis and different conventions used in the literature. Our basis is chosen to align with Ref. [3] where possible, and we write ——— for the two CP-even operators that were excluded from that work.

| | aQGC Operator Basis | C P | Almeida, Éboli, Gonzalez-Garcia [3] | Remmen & Rodd [4] | Murphy [6] |
|---|---|---|---|---|---|
| $\mathcal{O}_0^S$ | $[D_\mu H^\dagger D_\nu H][D^\mu H^\dagger D^\nu H]$ | + + | $\mathcal{O}_{S,0}$ | $\mathcal{O}_2^{H^4}$ | $Q_{H^4}^{(2)}$ |
| $\mathcal{O}_1^S$ | $[D^\mu H^\dagger D_\mu H][D^\nu H^\dagger D_\nu H]$ | + + | $\mathcal{O}_{S,1}$ | $\mathcal{O}_3^{H^4}$ | $Q_{H^4}^{(3)}$ |
| $\mathcal{O}_2^S$ | $[D_\mu H^\dagger D_\nu H][D^\nu H^\dagger D^\mu H]$ | + + | $\mathcal{O}_{S,2}$ | $\mathcal{O}_1^{H^4}$ | $Q_{H^4}^{(1)}$ |
| $\mathcal{O}_0^M$ | $\frac{1}{2}[D^\mu H^\dagger D_\mu H]W_{\nu\rho}^I W^{I\,\nu\rho}$ | + + | $\mathcal{O}_{M,0}$ | $\frac{1}{2}\mathcal{O}_2^{H^2W^2}$ | $\frac{1}{2}Q_{W^2H^2D^2}^{(2)}$ |
| $\mathcal{O}_1^M$ | $-\frac{1}{2}[D^\mu H^\dagger D^\nu H]W_{\mu\rho}^I W_\nu^{I\,\rho}$ | + + | $\mathcal{O}_{M,1}$ | $-\frac{1}{2}\mathcal{O}_1^{H^2W^2}$ | $-\frac{1}{2}Q_{W^2H^2D^2}^{(1)}$ |
| $\mathcal{O}_2^M$ | $[D^\mu H^\dagger D_\mu H]B_{\nu\rho}B^{\nu\rho}$ | + + | $\mathcal{O}_{M,2}$ | $\mathcal{O}_2^{H^2B^2}$ | $Q_{B^2H^2D^2}^{(2)}$ |
| $\mathcal{O}_3^M$ | $-[D^\mu H^\dagger D^\nu H]B_{\mu\rho}B_\nu^{\,\rho}$ | + + | $\mathcal{O}_{M,3}$ | $-\mathcal{O}_1^{H^2B^2}$ | $-Q_{B^2H^2D^2}^{(1)}$ |
| $\mathcal{O}_4^M$ | $[D^\mu H^\dagger \tau^I D_\mu H]B^{\nu\rho}W_{\nu\rho}^I$ | + + | $\mathcal{O}_{M,4}$ | $\mathcal{O}_1^{H^2BW}$ | $Q_{WBH^2D^2}^{(1)}$ |
| $\mathcal{O}_5^M$ | $[D^\mu H^\dagger \tau^I D^\nu H](B_\mu^{\,\rho}W_{\nu\rho}^I + B_\nu^{\,\rho}W_{\mu\rho}^I)$ | + + | $\mathcal{O}_{M,5}$ | $\mathcal{O}_3^{H^2BW}$ | $Q_{WBH^2D^2}^{(4)}$ |
| $\mathcal{O}_7^M$ | $[D^\mu H^\dagger \tau^I \tau^J D^\nu H]W_{\mu\rho}^J W_\nu^{I\,\rho}$ | + + | $\mathcal{O}_{M,7}$ | $\frac{1}{4}\mathcal{O}_1^{H^2W^2}-\frac{1}{2}\mathcal{O}_3^{H^2W^2}$ | $\frac{1}{4}Q_{W^2H^2D^2}^{(1)}-\frac{1}{2}Q_{W^2H^2D^2}^{(4)}$ |
| $\mathcal{O}_8^M$ | $i[D^\mu H^\dagger \tau^I D^\nu H](B_\mu^{\,\rho}\widetilde{W}_{\nu\rho}^I - B_\nu^{\,\rho}\widetilde{W}_{\mu\rho}^I)$ | − − | ——— | $\widetilde{\mathcal{O}}_2^{H^2BW}$ | $Q_{WBH^2D^2}^{(5)}$ |
| $\mathcal{O}_9^M$ | $\epsilon^{IJK}[D^\mu H^\dagger \tau^I D^\nu H](W_{\mu\rho}^J \widetilde{W}_\nu^{K\,\rho} - \widetilde{W}_{\mu\rho}^J W_\nu^{K\,\rho})$ | − − | ——— | $\widetilde{\mathcal{O}}_2^{H^2W^2}$ | $Q_{W^2H^2D^2}^{(5)}$ |
| $\mathcal{O}_0^T$ | $\frac{1}{4}W_{\mu\nu}^I W^{I\,\mu\nu}W_{\rho\sigma}^J W^{J\,\rho\sigma}$ | + + | $\mathcal{O}_{T,0}$ | $\frac{1}{4}\mathcal{O}_1^{W^4}$ | $\frac{1}{4}Q_{W^4}^{(1)}$ |
| $\mathcal{O}_1^T$ | $\frac{1}{4}W_{\mu\nu}^I W^{J\,\mu\nu}W_{\rho\sigma}^I W^{J\,\rho\sigma}$ | + + | $\mathcal{O}_{T,1}$ | $\frac{1}{4}\mathcal{O}_3^{W^4}$ | $\frac{1}{4}Q_{W^4}^{(3)}$ |
| $\mathcal{O}_2^T$ | $\frac{1}{4}W_{\mu\nu}^I W^{I\,\nu\alpha}W_{\alpha\beta}^J W^{J\,\beta\mu}$ | + + | $\mathcal{O}_{T,2}$ | $\frac{1}{16}\mathcal{O}_1^{W^4}+\frac{1}{16}\mathcal{O}_3^{W^4}+\frac{1}{16}\mathcal{O}_4^{W^4}$ | $\frac{1}{16}Q_{W^4}^{(1)}+\frac{1}{16}Q_{W^4}^{(3)}+\frac{1}{16}Q_{W^4}^{(4)}$ |
| $\mathcal{O}_3^T$ | $\frac{1}{4}W_{\mu\nu}^I W^{J\,\nu\alpha}W_{\alpha\beta}^I W^{J\,\beta\mu}$ | + + | $\mathcal{O}_{T,3}$ | $\frac{1}{8}\mathcal{O}_3^{W^4}+\frac{1}{16}\mathcal{O}_2^{W^4}$ | $\frac{1}{8}Q_{W^4}^{(3)}+\frac{1}{16}Q_{W^4}^{(2)}$ |
| $\mathcal{O}_4^T$ | $\frac{1}{2}W_{\mu\nu}^I B^{\nu\alpha}W_{\alpha\beta}^I B^{\beta\mu}$ | + + | $\mathcal{O}_{T,4}$ | $\frac{1}{8}\mathcal{O}_2^{B^2W^2}+\frac{1}{4}\mathcal{O}_3^{B^2W^2}$ | $\frac{1}{8}Q_{W^2B^2}^{(2)}+\frac{1}{4}Q_{W^2B^2}^{(3)}$ |
| $\mathcal{O}_5^T$ | $\frac{1}{2}B_{\mu\nu}B^{\mu\nu}W_{\rho\sigma}^I W^{I\,\rho\sigma}$ | + + | $\mathcal{O}_{T,5}$ | $\frac{1}{2}\mathcal{O}_2^{B^2W^2}$ | $\frac{1}{2}Q_{W^2B^2}^{(1)}$ |
| $\mathcal{O}_6^T$ | $\frac{1}{2}B_{\mu\nu}W^{I\,\mu\nu}B_{\rho\sigma}W^{I\,\rho\sigma}$ | + + | $\mathcal{O}_{T,6}$ | $\frac{1}{2}\mathcal{O}_3^{B^2W^2}$ | $\frac{1}{2}Q_{W^2B^2}^{(3)}$ |
| $\mathcal{O}_7^T$ | $\frac{1}{2}W_{\mu\nu}^I W^{I\,\nu\alpha}B_{\alpha\beta}B^{\beta\mu}$ | + + | $\mathcal{O}_{T,7}$ | $\frac{1}{8}\mathcal{O}_1^{B^2W^2}+\frac{1}{8}\mathcal{O}_3^{B^2W^2}+\frac{1}{8}\mathcal{O}_4^{B^2W^2}$ | $\frac{1}{8}Q_{W^2B^2}^{(1)}+\frac{1}{8}Q_{W^2B^2}^{(3)}+\frac{1}{8}Q_{W^2B^2}^{(4)}$ |
| $\mathcal{O}_8^T$ | $B_{\mu\nu}B^{\mu\nu}B_{\rho\sigma}B^{\rho\sigma}$ | + + | $\mathcal{O}_{T,8}$ | $\mathcal{O}_1^{B^4}$ | $Q_{B^4}^{(1)}$ |
| $\mathcal{O}_9^T$ | $B_{\mu\nu}B^{\nu\alpha}B_{\alpha\beta}B^{\beta\mu}$ | + + | $\mathcal{O}_{T,9}$ | $\frac{1}{2}\mathcal{O}_1^{B^4}+\frac{1}{4}\mathcal{O}_2^{B^4}$ | $\frac{1}{2}Q_{B^4}^{(1)}+\frac{1}{4}Q_{B^4}^{(2)}$ |

hypercharge field strength. Accordingly, the parity transformation of an operator is simply controlled by the number of dual field strength tensors.

Charge conjugation is more subtle. Our description of CP transformations follows Ref. [5] (correcting minor sign issues noticed in Eqs. (4.1) and (4.3) of that work). Let us consider fields transforming under an SU($N$) gauge group. The definition of charge conjugation then involves a matrix $C$ acting on fundamental indices, which is unitary so $C^\dagger C = \mathbb{I}$ and satisfies $CC^* = \pm\mathbb{I}$. For $N$ odd, only the plus sign is allowed in the latter equality. A fundamental representation can then be defined to transform as C: $H \to CH^*$. For consistency, an adjoint representation (analogous to an $HH^\dagger$ combination of fundamentals) then transforms as C: $W \to -CW^T C^\dagger$, where the overall phase should be just a sign for a real representation and should be a minus sign to preserve the Lie algebra. In the Abelian case, the above gauge-field transformation reduces to C: $B \to -B$.

A combination like $H_a^\dagger W_1...W_n H_b$, which arises in M-type operators, then transforms as:

$$\text{C}: H_a^\dagger W_1...W_n H_b \to (H_a^T C^\dagger)(-CW_1^T C^\dagger)...(-CW_n^T C^\dagger)(CH_b^*) \tag{2}$$
$$= (-1)^n H_b^\dagger W_n...W_1 H_a \,.$$

Table 2: As in Table 1, but for the CP-odd aQGC operators. We write "N/A" for the conversion to Ref. [3] as that work did not consider CP-odd operators.

| | aQGC Operator Basis | C | P | AEG [3] | RR [4] | M [6] |
|---|---|---|---|---|---|---|
| $\mathscr{O}_1^M$ | $[D^\mu H^\dagger D_\mu H]B_{\nu\rho}\widetilde{B}^{\nu\rho}$ | $+$ | $-$ | N/A | $\widetilde{\mathcal{O}}_1^{H^2B^2}$ | $Q_{B^2H^2D^2}^{(3)}$ |
| $\mathscr{O}_2^M$ | $[D^\mu H^\dagger \tau^I D_\mu H]B_{\nu\rho}\widetilde{W}^{I\,\nu\rho}$ | $+$ | $-$ | N/A | $\widetilde{\mathcal{O}}_1^{H^2BW}$ | $\frac{1}{2}Q_{WBH^2D^2}^{(2)}$ |
| $\mathscr{O}_3^M$ | $i[D^\mu H^\dagger \tau^I D^\nu H](B_{\mu\rho}W_\nu^{I\,\rho}-B_{\nu\rho}W_\mu^{I\,\rho})$ | $-$ | $+$ | N/A | $\mathcal{O}_2^{H^2BW}$ | $\frac{1}{2}Q_{WBH^2D^2}^{(3)}$ |
| $\mathscr{O}_4^M$ | $[D^\mu H^\dagger \tau^I D^\nu H](B_{\mu\rho}\widetilde{W}_\nu^{I\,\rho}+B_{\nu\rho}\widetilde{W}_\mu^{I\,\rho})$ | $+$ | $-$ | N/A | $\widetilde{\mathcal{O}}_3^{H^2BW}$ | $\frac{1}{2}Q_{WBH^2D^2}^{(6)}$ |
| $\mathscr{O}_5^M$ | $[D^\mu H^\dagger D_\mu H]W_{\nu\rho}^I\widetilde{W}^{I\,\nu\rho}$ | $+$ | $-$ | N/A | $\widetilde{\mathcal{O}}_1^{H^2W^2}$ | $Q_{W^2H^2D^2}^{(3)}$ |
| $\mathscr{O}_6^M$ | $i\,\epsilon^{IJK}[D^\mu H^\dagger \tau^I D^\nu H](W_{\mu\rho}^J\widetilde{W}_\nu^{K\,\rho}+\widetilde{W}_{\mu\rho}^J W_\nu^{K\,\rho})$ | $+$ | $-$ | N/A | $\widetilde{\mathcal{O}}_3^{H^2W^2}$ | $\frac{1}{2}Q_{W^2H^2D^2}^{(6)}$ |
| $\mathscr{O}_1^T$ | $B_{\mu\nu}B^{\mu\nu}B_{\rho\sigma}\widetilde{B}^{\rho\sigma}$ | $+$ | $-$ | N/A | $\widetilde{\mathcal{O}}_1^{B^4}$ | $Q_{B^4}^{(3)}$ |
| $\mathscr{O}_2^T$ | $B_{\mu\nu}\widetilde{B}^{\mu\nu}W_{\rho\sigma}^I W^{I\rho\sigma}$ | $+$ | $-$ | N/A | $\widetilde{\mathcal{O}}_1^{B^2W^2}$ | $Q_{W^2B^2}^{(5)}$ |
| $\mathscr{O}_3^T$ | $B_{\mu\nu}B^{\mu\nu}W_{\rho\sigma}^I\widetilde{W}^{I\rho\sigma}$ | $+$ | $-$ | N/A | $\widetilde{\mathcal{O}}_2^{B^2W^2}$ | $Q_{W^2B^2}^{(6)}$ |
| $\mathscr{O}_4^T$ | $B_{\mu\nu}W^{I\,\mu\nu}B_{\rho\sigma}\widetilde{W}^{I\rho\sigma}$ | $+$ | $-$ | N/A | $\widetilde{\mathcal{O}}_3^{B^2W^2}$ | $Q_{W^2B^2}^{(7)}$ |
| $\mathscr{O}_5^T$ | $W_{\mu\nu}^I W^{I\,\mu\nu}W_{\rho\sigma}^J\widetilde{W}^{J\rho\sigma}$ | $+$ | $-$ | N/A | $\widetilde{\mathcal{O}}_1^{W^4}$ | $Q_{W^4}^{(5)}$ |
| $\mathscr{O}_6^T$ | $W_{\mu\nu}^I W^{J\,\mu\nu}W_{\rho\sigma}^I\widetilde{W}^{J\rho\sigma}$ | $+$ | $-$ | N/A | $\widetilde{\mathcal{O}}_2^{W^4}$ | $Q_{W^4}^{(6)}$ |

We thus see that charge conjugation reverses the order of the fields and introduces an extra minus sign for odd numbers of adjoint representations. For instance, the $\mathcal{O}_9^M$ and $\mathscr{O}_6^M$ operators involve two terms of the form $(D_\mu H)^\dagger[W_{\mu\rho},\widetilde{W}_{\nu\rho}](D_\nu H)$, which transform under charge conjugation as

$$\mathrm{C}:(D_\mu H)^\dagger[W_{\mu\rho},\widetilde{W}_{\nu\rho}](D_\nu H)\to(D_\nu H)^\dagger[\widetilde{W}_{\nu\rho},W_{\mu\rho}](D_\mu H) \tag{3}$$
$$=(D_\mu H)^\dagger[\widetilde{W}_{\mu\rho},W_{\nu\rho}](D_\nu H)\,,$$

where the last equality only involves a relabeling of the Lorentz indices. In such a combination of fields, the adjoint field strength and its dual are therefore just exchanged (leaving all indices untouched). Since $\mathcal{O}_9^M$ is by construction odd under this exchange, it is therefore odd under charge conjugation. Given that $\mathcal{O}_9^M$ is also odd under parity (because of the dual field strength), it is actually CP-even. On the contrary, $\mathscr{O}_6^M$ is even under the exchange of the field strength and its dual. It is therefore even under charge conjugation, while also being parity odd, so it is CP-odd altogether.

There are two specific realizations of the $C$ matrix often employed in the literature (see again Ref. [5]): one symmetric $C_S=\mathbb{I}$, and one skew $C_A=i\sigma_2$. The latter satisfies $C_A C_A^*=-\mathbb{I}$ and is allowed for SU($N$) with $N=2$ even as in the electroweak sector of the SM. With these specific representations, the transformation of various field components and combinations are the following:

$$
\begin{aligned}
C_S &: H\to H^*\,, & C_A &: H\to(i\sigma_2)H^*\,,\\
C_S &: B_{\mu\nu}\to -B_{\mu\nu}\,, & C_A &: B_{\mu\nu}\to -B_{\mu\nu}\,,\\
C_S &: W_{\mu\nu}^I\to f(1-\delta_{I2})W_{\mu\nu}^I\,, & C_A &: W_{\mu\nu}^I\to +W_{\mu\nu}^I\,,\\
C_S &: (D^\mu H^\dagger \tau^I D^\nu H)\to f(\delta_{I2})(D^\nu H^\dagger \tau^I D^\mu H)\,, & C_A &: (D^\mu H^\dagger \tau^I D^\nu H)\to -(D^\nu H^\dagger \tau^I D^\mu H)\,,
\end{aligned}
\tag{4}
$$

where the final result on the last line can be derived from that of the first line. Note that this final expression involves an interchange $\mu \leftrightarrow \nu$ in addition to the prefactor (which cancels out in both cases in a $(D^\mu H^\dagger \tau^I D^\nu H) W^I_{\rho\sigma}$ combination).

# 4 Monte Carlo implementation

An implementation of the C-even and P-even operators of Refs. [1–3] in a UFO model enabling event generation in various Monte Carlo simulation tools is available at https://feynrules.irmp.ucl.ac.be/wiki/AnomalousGaugeCoupling. The code provided by the authors of Ref. [9] at https://www.fuw.edu.pl/smeft also allows one to generate a UFO model implementation of the aQGC operators following the conventions of Ref. [6] (see also the conversion between the two bases provided at https://www.fuw.edu.pl/smeft/Validation.pdf). We provide an implementation of all the dimension-eight aQGC operators listed in Table 1 and Table 2 at https://github.com/gdurieux/aqgc. Adopting the same parameter naming as in AnomalousGaugeCoupling whenever possible, we label the coefficients of $\mathcal{O}_i^{S,M,T}$ operators as FSi, FMi, and FTi. To simplify notation, the coefficients of the operators in the bases of Refs. [4,6] (also implemented for completeness) are relabeled as $c_i^{S,M,T}$ below and appear as cSi, cMi, and cTi in the UFO model implementation. Their definitions are:

$$
\begin{aligned}
& c_1^M \equiv c_1^{H^2B^2} = c_{B^2H^2D^2}^{(1)}, && c_1^T \equiv c_1^{B^4} = c_{B^4}^{(1)}, \\
& c_2^M \equiv c_2^{H^2B^2} = c_{B^2H^2D^2}^{(2)}, && c_2^T \equiv c_2^{B^4} = c_{B^4}^{(2)}, \\
& c_3^M \equiv c_1^{H^2BW} = 2c_{WBH^2D^2}^{(1)}, && c_3^T \equiv c_1^{B^2W^2} = c_{W^2B^2}^{(1)}, \\
c_1^S \equiv c_1^{H^4} = c_{H^4}^{(1)}, \quad & c_4^M \equiv \tilde{c}_2^{H^2BW} = 2c_{WBH^2D^2}^{(5)}, && c_4^T \equiv c_2^{B^2W^2} = c_{W^2B^2}^{(2)}, \\
c_2^S \equiv c_2^{H^4} = c_{H^4}^{(2)}, \quad & c_5^M \equiv c_3^{H^2BW} = 2c_{WBH^2D^2}^{(4)}, && c_5^T \equiv c_3^{B^2W^2} = c_{W^2B^2}^{(3)}, \\
c_3^S \equiv c_3^{H^4} = c_{H^4}^{(3)}, \quad & c_6^M \equiv c_1^{H^2W^2} = c_{W^2H^2D^2}^{(1)}, && c_6^T \equiv c_4^{B^2W^2} = c_{W^2B^2}^{(4)}, \\
& c_7^M \equiv c_2^{H^2W^2} = c_{W^2H^2D^2}^{(2)}, && c_7^T \equiv c_1^{W^4} = c_{W^4}^{(1)}, \\
& c_8^M \equiv c_3^{H^2W^2} = 2c_{W^2H^2D^2}^{(4)}, && c_8^T \equiv c_2^{W^4} = c_{W^4}^{(2)}, \\
& c_9^M \equiv \tilde{c}_2^{H^2W^2} = 2c_{W^2H^2D^2}^{(5)}, && c_9^T \equiv c_3^{W^4} = c_{W^4}^{(3)}, \\
& && c_{10}^T \equiv c_4^{W^4} = c_{W^4}^{(4)}.
\end{aligned} \tag{5}
$$

The correspondence between the two sets of coefficients derives directly from the map between operators provided in Table 1:

$$
\begin{aligned}
& && FT0 = 4\,cT7 - 4\,cT10, \\
& FM0 = 2\,cM7, && FT1 = -8\,cT8 + 4\,cT9 - 4\,cT10, \\
& FM1 = -2\,cM6 - cM8, && FT2 = 16\,cT10, \\
& FM2 = cM2, && FT3 = 16\,cT8, \\
FS0 = cS2, \quad & FM3 = -cM1, && FT4 = 8\,cT4, \\
FS1 = cS3, \quad & FM4 = cM3, && FT5 = 2\,cT3 - 2\,cT6, \\
FS2 = cS1, \quad & FM5 = cM5, && FT6 = -4\,cT4 + 2\,cT5 - 2\,cT6, \\
& FM7 = -2\,cM8, && FT7 = 8\,cT6, \\
& FM8 = cM4, && FT8 = cT1 - 2\,cT2, \\
& FM9 = cM9, && FT9 = 4\,cT2.
\end{aligned} \tag{6}
$$

Although a similar naming scheme is used, beware that FSi, FMi, FTi are the coefficients of the $\mathcal{O}_i^{S,M,T}$ operators, not cSi, cMi, cTi.

## 5 Massless amplitudes

Considering massless amplitudes instead of operators, one can readily form linear combinations with definite C and P transformation properties. This provides an alternative view on the independent aQGC gauge and kinematic structures. Table 3 lists these independent linear combinations. They are symmetric under the exchange of identical bosons as required by Bose statistics. Momenta and gauge indices are all substituted by the corresponding particle labels. Parity just exchanges square and angle spinors, which is equivalent to a complex conjugation of the kinematic structures. Charge conjugation effectively acts as a complex conjugation of the gauge structure, in combination with an exchange of conjugate particle labels. It therefore exchanges the $t$ and $u$ Mandelstam invariants in various amplitudes of Table 3. The only gauge structure that is effectively C-odd is the anticommutator $[\tau^1, \tau^2]^4_3$, appearing in $W_1 W_2 H_3^* H_4$ amplitudes, that is sent to its Hermitian conjugate according to Eq. (2).

## 6 Outlook

In the absence of unambiguous signs of new light states, SMEFT is perhaps our best tool for constraining and characterizing heavy new physics using measurements at currently accessible scales. Interestingly, the coverage of the SMEFT operator coefficient space by healthy ultraviolet completions is not uniform: certain signs and magnitudes of operator coefficients can be forbidden, irrespective of the details of new physics, using general axioms of quantum field theory, specifically unitarity, causality, and locality [10]. These principles, and in particular their consequences for the analytic properties of forward scattering amplitudes, lead to "positivity bounds" on the SMEFT coefficients, which are particularly powerful at mass dimension eight, including the aQGCs (see, e.g., Ref. [4,11]). As the LHC collaborations are placing constraints on aQGC coefficients, facilitating the use of positivity constraints would be desirable. A first step in this direction was made here, by providing a complete reference basis of aQGC operators.

## Acknowledgments

We thank Patrick Dougan, Ilaria Brivio, Hannes Mildner, Matteo Presilla, Marc Riembau, Despoina Sampsonidou, and Dave Sutherland for comments.

**Funding information**    GD is a Research Associate of the Fund for Scientific Research – FNRS, Belgium. GNR is supported by the James Arthur Postdoctoral Fellowship at New York University. The work of NLR was supported by the Office of High Energy Physics of the U.S. Department of Energy under contract DE-AC02-05CH11231.

Table 3: Independent linear combinations of massless dimension-eight amplitudes involving four electroweak bosons. The CP-even coefficients are given in terms of the shorthand notation of Eq. (5).

| particles | gauge structure | kinematics | C | P | coefficient |
|---|---|---|---|---|---|
| $H_1^* H_2^* H_3 H_4$ | $\delta_1^3\delta_2^4 + \delta_1^4\delta_2^3$ | $s^2$ | + | + | $\frac{i}{2}c_2^S$ |
| | $\delta_1^3\delta_2^4 + \delta_1^4\delta_2^3$ | $t^2 + u^2$ | + | + | $\frac{i}{4}(c_1^S + c_3^S)$ |
| | $\delta_1^3\delta_2^4 - \delta_1^4\delta_2^3$ | $t^2 - u^2$ | + | + | $-\frac{i}{4}(c_1^S - c_3^S)$ |
| $B_1 B_2 H_3^* H_4$ | | $([12]^2 + \langle 12\rangle^2)\,s$ | + | + | $\frac{i}{4}(c_1^M + 4c_2^M)$ |
| | | $([12]^2 - \langle 12\rangle^2)\,s$ | + | − | $-\not{c}_1^M$ |
| | | $[1(3-4)2]^2 + \langle 1(3-4)2\rangle^2$ | + | + | $-\frac{i}{8}c_1^M$ |
| $W_1 B_2 H_3^* H_4$ | $[\tau^1]_3^4$ | $([12]^2 + \langle 12\rangle^2)\,s$ | + | + | $\frac{i}{4}(2c_3^M + c_5^M)$ |
| | $[\tau^1]_3^4$ | $([12]^2 - \langle 12\rangle^2)\,s$ | + | − | $-\frac{1}{4}(2\not{c}_2^M + \not{c}_4^M)$ |
| | $[\tau^1]_3^4$ | $([12]^2 + \langle 12\rangle^2)(t-u)$ | − | + | $-\frac{1}{4}\not{c}_3^M$ |
| | $[\tau^1]_3^4$ | $([12]^2 - \langle 12\rangle^2)(t-u)$ | − | − | $-\frac{i}{4}c_4^M$ |
| | $[\tau^1]_3^4$ | $[1(3-4)2]^2 + \langle 1(3-4)2\rangle^2$ | + | + | $-\frac{i}{8}c_5^M$ |
| | $[\tau^1]_3^4$ | $[1(3-4)2]^2 - \langle 1(3-4)2\rangle^2$ | + | − | $\frac{1}{8}\not{c}_4^M$ |
| $W_1 W_2 H_3^* H_4$ | $\delta^{12}\delta_3^4$ | $([12]^2 + \langle 12\rangle^2)\,s$ | + | + | $\frac{i}{4}(c_6^M + 4c_7^M)$ |
| | $\delta^{12}\delta_3^4$ | $([12]^2 - \langle 12\rangle^2)\,s$ | + | − | $-\not{c}_5^M$ |
| | $[\tau^1,\tau^2]_3^4$ | $([12]^2 + \langle 12\rangle^2)(t-u)$ | + | + | $\frac{i}{4}c_8^M$ |
| | $[\tau^1,\tau^2]_3^4$ | $([12]^2 - \langle 12\rangle^2)(t-u)$ | + | − | $-\frac{1}{2}\not{c}_6^M$ |
| | $\delta^{12}\delta_3^4$ | $[1(3-4)2]^2 + \langle 1(3-4)2\rangle^2$ | + | + | $-\frac{i}{8}c_6^M$ |
| | $[\tau^1,\tau^2]_3^4$ | $[1(3-4)2]^2 - \langle 1(3-4)2\rangle^2$ | − | − | $\frac{i}{4}c_9^M$ |
| $B_1 B_2 B_3 B_4$ | | $([12]^2[34]^2 + \langle 12\rangle^2\langle 34\rangle^2) + \text{perm.}$ | + | + | $8i(c_1^T - c_2^T)$ |
| | | $([12]^2[34]^2 - \langle 12\rangle^2\langle 34\rangle^2) + \text{perm.}$ | + | − | $-8\not{c}_1^T$ |
| | | $([12]^2\langle 34\rangle^2 + \langle 12\rangle^2[34]^2) + \text{perm.}$ | + | + | $8i(c_1^T + c_2^T)$ |
| $B_1 B_2 W_3 W_4$ | $\delta^{34}$ | $[12]^2[34]^2 + \langle 12\rangle^2\langle 34\rangle^2$ | + | + | $4i(c_3^T - c_4^T)$ |
| | $\delta^{34}$ | $[12]^2[34]^2 - \langle 12\rangle^2\langle 34\rangle^2$ | + | − | $-4(c_2^T + c_3^T)$ |
| | $\delta^{34}$ | $([13]^2[24]^2 + [14]^2[23]^2) + (\langle 13\rangle^2\langle 24\rangle^2 + \langle 14\rangle^2\langle 23\rangle^2)$ | + | + | $2i(c_5^T - c_6^T)$ |
| | $\delta^{34}$ | $([13]^2[24]^2 + [14]^2[23]^2) - (\langle 13\rangle^2\langle 24\rangle^2 + \langle 14\rangle^2\langle 23\rangle^2)$ | + | − | $-2\not{c}_4^T$ |
| | $\delta^{34}$ | $[12]^2\langle 34\rangle^2 + \langle 12\rangle^2[34]^2$ | + | + | $4i(c_3^T + c_4^T)$ |
| | $\delta^{34}$ | $[12]^2\langle 34\rangle^2 - \langle 12\rangle^2[34]^2$ | + | − | $-4(\not{c}_2^T - \not{c}_3^T)$ |
| | $\delta^{34}$ | $[13]^2\langle 24\rangle^2 + [14]^2\langle 23\rangle + \langle 13\rangle^2[24]^2 + \langle 14\rangle^2[23]^2$ | + | + | $2i(c_5^T + c_6^T)$ |
| $W_1 W_2 W_3 W_4$ | $\delta^{12}\delta^{34} + \text{perm.}$ | $([12]^2[34]^2 + \langle 12\rangle^2\langle 34\rangle^2) + \text{perm.}$ | + | + | $4i(c_9^T - c_{10}^T)$ |
| | $\delta^{12}\delta^{34} + \text{perm.}$ | $([12]^2[34]^2 - \langle 12\rangle^2\langle 34\rangle^2) + \text{perm.}$ | + | − | $-4\not{c}_6^T$ |
| | $\{\delta^{12}\delta^{34}$ | $([12]^2[34]^2 + \langle 12\rangle^2\langle 34\rangle^2)\} + \text{perm.}$ | + | + | $4i(2c_7^T - 2c_8^T - c_9^T + c_{10}^T)$ |
| | $\{\delta^{12}\delta^{34}$ | $([12]^2[34]^2 - \langle 12\rangle^2\langle 34\rangle^2)\} + \text{perm.}$ | + | − | $-4(2\not{c}_5^T - \not{c}_6^T)$ |
| | $\delta^{12}\delta^{34} + \text{perm.}$ | $([12]^2\langle 34\rangle^2 + \langle 12\rangle^2[34]^2) + \text{perm.}$ | + | + | $4i(c_9^T + c_{10}^T)$ |
| | $\{\delta^{12}\delta^{34}$ | $([12]^2\langle 34\rangle^2 + \langle 12\rangle^2[34]^2)\} + \text{perm.}$ | + | + | $4(2c_7^T + 2c_8^T - c_9^T - c_{10}^T)$ |

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
