# Peer review of "LHC EFT WG Note: Basis for Anomalous Quartic Gauge Couplings"

_SciPost Physics Community Reports, doi:SciPost Phys. Comm. Rep. 6 (2025)_

## Round 1 · Referee Report · Anonymous (Referee 1) · 2024-12-13

Report
The report presents a community effort to clarify - once and for all - the composition and properties of the basis of dimension 8 SMEFT operators that generate anomalous Quartic Gauge Couplings (aQGC), but no anomalous Triple Gauge Couplings (aTGC). The phenomenology of such operators has been studied for a long time, and the ATLAS and CMS collaborations have been long using this formalism to search for anomalies in their measurements, in particular of Vector Boson Scattering processes.
The basis employed was corrected several times in the past years, and the CP properties were never established in a fully correct way, leading to multiple inconvenient changes of notation and updates of the corresponding Monte Carlo tools. The results of this report, which is co-authored by members of all groups who have previously attempted to close the deal on the operator set, are therefore very welcome.
There is no doubt that this report fulfills all requirements for publication in this journal.
I did my best to cross-check the formulas presented, as the value of the report clearly hinges on them being correct and free from typesetting errors.
I only found a few minor points, that I kindly ask the authors to look into (hopefully one last time) before recommending the report for publication.
1- is a factor 1/2 missing in the definition of $O^M_4, O^M_5$?
ref [3] uses $\widehat{W}{\mu\nu} = \frac{W^i$} \tau^i }{2.
in $O^M_1$, for instance, the 1/4 coming from the two W partially cancels with a 2 from $tr(\tau^i \tau^j) = 2 \delta^{ij}$, but in $O^M_4, O^M_5$ there are no traces of the pauli matrices, so I'm not seeing how the 1/2 could go away
2- for the same reason, is a 1/4 missing in the definition of $O^M_7$?
3- of all operators, I couldn't understand the conversion of $O^M_7$ to the notation of refs [4,6].
using the pauli matrix relation $\tau^i \tau^j = \delta^{ij} + i \epsilon^{ijk} \tau^k$ and keeping the current normalization in the definition of $O^M_7$, I find:
$O^M_7 = O_1 ^{H^2W^2} - O_3^{H^2W^2}$
where the minus sign in the second term comes from relabeling $ijk\to kji$ and then taking $\epsilon^{kji}=-\epsilon^{ijk}$.
I'm not sure where the relative factor 2 could come from.
4- as a minor comment, for the sake of having all conventions spelled out, it would be useful to indicate the conventions adopted for the definition of dual field strength and for the covariant derivative sign in the monte carlo codes.
Recommendation
Ask for minor revision
We thank the referee for going over our work so carefully and double checking even fine details of the bases. Let us respond to the comments raised. We believe there is actually a common issue that resolves the first three of the referee's queries. In particular, we did not define our convention for the basis of the generators. Following our Ref. [4], we took \(\tau^I = \sigma^I/2\), with \(\sigma^I\) the Pauli matrices. That therefore means we have in the notation defined below Eq. (3) of our Ref. [3], With that convention, our operator definitions exactly match Ref. [3] for \({\cal O}_{4,5}^M\). This is of course a key point, however, and we will add an explicit definition of \(\tau^I\) to the end of section 1. We believe the factor of 1/4 the referee asked about in \({\cal O}_7^M\) is resolved identically to the 1/2 discussed above. We evaluated the mapping between the two bases as follows. This appears to be the exact same manipulation made by the referee up to the factors of 1/4 and 1/2 which again arise from \(\tau^I = \sigma^I/2\). We will explicitly define the dual field strengths and our convention for the covariant derivative in section 1 also.
I thank the authors for their reply.
Indeed, the conventions on the definition of $\tau^I$ were unclear, and taking $\tau^I=\sigma^I/2$ solves all my queries about basis conversion.
There is only one issue left, which is that Ref [6] (Murphy, see below their Eq. 2.2) defines $\tau^I=\sigma^I$, which means there should be factors 2 between the definitions in [4] and [6] for all operators with explicit $\tau$'s.
Would the authors agree?
We had completely missed this different use of $\tau^I$ in the Murphy basis and thank the referee for bringing it to our attention. This will introduce additional factors of $1/2$ in the mapping we employ for several of the CP-even and odd mixed type operators, but we will implement these carefully in the updated manuscript. In particular, the operators $Q_{W^2H^2D^2}^{4,5,6}$ and $Q_{W BH^2D^2}^{1-6}$ in the Murphy notation will be updated with an additional factor of $1/2$.

Author: Nicholas Rodd on 2025-01-07 [id 5094]
(in reply to Report 2 on 2025-01-02)We thank the referee for their careful reading of the manuscript and FeynRules implementation. We respond to their four comments below.

---

## Round 1 · Referee Report · Anonymous (Referee 2) · 2025-1-2

Report
In this article, the authors present the first complete basis of the dimension-eight SMEFT operators that induce anomalous quartic gauge couplings (aQGC) in the electroweak sector. This corrects several previous, albeit incomplete, efforts by various groups (including some of the authors) to establish such a basis. The article also provides the relations of the presented basis to the ones previously used in the literature. Thus, this paper establishes a valuable foundation for the community to explore aQGC.
The paper is well written and clearly formulated. I only have a few suggestions for improvements:
1) I believe the introduction would benefit from a brief discussion of why aQGC are interesting to study.
2) It is very useful, that the authors also provide a FeynRules implementation of their operator basis in a public GitHub repository that is linked in section 4. The relations of the corresponding operator coefficients are given in Eq. (5). The notation and naming convention are, however, not fully clear when only looking at the paper itself. I would therefore suggest to either properly define the notation and give a few more detail on what $\texttt{FS0},\ldots$ and $\texttt{cS1},\ldots$ exactly correspond to, or to possibly move this equation entirely to the GitHub repository, where all coefficients are defined in the FeynRules file. In its current form, Eq. (5) is difficult to understand if one is not looking at the repository in parallel.
3) The notation chosen for the Wilson coefficients in Eq. (6) is rather confusing. While $c_1^{H^2B^2}$ is the coefficient of $\mathcal{O}_1^{H^2B^2}$ and $c_{ B^2 H^2D^2}^{(1)}$ is the coefficient of $\mathcal{O}_{ B^2 H^2D^2}^{(1)}$, it appears that the coefficient of $\mathcal{O}_{3}^M$ is not given by $c_{3}^M$ but instead by $-c_{1}^M$, and similar naming issues for the other operators. I would thus suggest using a different notation for the coefficients appearing on the left-hand side of Eq. (6), or to at least clearly state the issues associated with the chosen notation.
4) There is a typo in the acknowledgements: “the the”.
Recommendation
Ask for minor revision

---

## Editorial Decision

published